# Can Grafting Manage Fusarium Wilt Disease of Cucumber and Increase Productivity under Heat Stress?

**DOI:** 10.3390/plants11091147

**Published:** 2022-04-24

**Authors:** Tarek A. Shalaby, Naglaa A. Taha, Mohamed T. Rakha, Hossam S. El-Beltagi, Wael F. Shehata, Khaled M. A. Ramadan, Hassan El-Ramady, Yousry A. Bayoumi

**Affiliations:** 1Arid Land Agriculture Department, College of Agricultural and Food Science, King Faisal University, P.O. Box 400, Al-Ahsa 31982, Saudi Arabia; 2Horticulture Department, Faculty of Agriculture, Kafrelsheikh University, Kafr El-Sheikh 33516, Egypt; mdrakha@gmail.com (M.T.R.); yousry.bayoumi@agr.kfs.edu.eg (Y.A.B.); 3Plant Pathology Research Institute, Agriculture Research Center, Giza 12619, Egypt; naglaa_abdelbaset@yahoo.com; 4Agricultural Biotechnology Department, College of Agricultural and Food Science, King Faisal University, P.O. Box 400, Al-Ahsa 31982, Saudi Arabia; wshehata@kfu.edu.sa; 5Biochemistry Department, Faculty of Agriculture, Cairo University, Giza 12613, Egypt; 6Plant Production Department, College of Environmental Agricultural Science, El–Arish University, North Sinai 45511, Egypt; 7Central Laboratories, Department of Chemistry, King Faisal University, Al-Ahsa 31982, Saudi Arabia; kramadan@kfu.edu.sa; 8Biochemistry Department, Faculty of Agriculture, Ain Shams University, Cairo 11566, Egypt; 9Soil and Water Department, Faculty of Agriculture, Kafrelsheikh University, Kafr El-Sheikh 33516, Egypt; hassan.elramady@agr.kfs.edu.eg; 10Institute of Animal Science, Biotechnology and Nature Conservation, Faculty of Agricultural and Food Sciences and Environmental Management, University of Debrecen, 138 Böszörményi Street, 4032 Debrecen, Hungary; 11Physiology & Breeding of Horticultural Crops Laboratory, Horticulture Department, Faculty of Agriculture, Kafrelsheikh University, Kafr El-Sheikh 33516, Egypt

**Keywords:** cucurbit rootstocks, anatomical structure, biotic and abiotic stress, disease incidence, disease severity, cucurbitaceous

## Abstract

Cucumber production is considered a crucial problem under biotic and abiotic stress, particularly in arid and semi-arid zones. The current study investigated the impact of grafted cucumber plants on five cucurbit rootstocks under infection with *Fusarium oxysporum* f. sp. *cucumerinum* alone and in combination with heat stress in two different locations (i.e., Kafr El-Sheikh and Sidi Salem) during the year of 2021. The rootstock of VSS-61 F_1_ displayed the highest level of resistance with values 20.8 and 16.6% for wilt incidence and 79.2 and 83.4% for the wilt reduction, respectively for both locations. This rootstock showed the lowest disease severity of fusarium wilt (15.3 and 12%), and high grafting efficiency (85 and 88%), respectively in both locations. Grafting also improved plant vigor and cucumber production under heat stress (40–43 °C). The rootstocks VSS-61 F_1_, Ferro and Super Shintoza significantly increased the total yield of cucumber plants compared to non-grafted cucumber and the rootstock Bottle gourd in both locations. Further studies are needed on grafted plants under multiple stresses in terms of plant biological levels, including physiological, biochemical and genetic attributes.

## 1. Introduction

Cucumber crop productivity mainly depends on the availability of natural resources, including soil, water, nutrients and air as well as climatic elements. This production is threatened by several constraints such as climate change and global warming, and abiotic and biotic stress [1]. The biotic stresses represent a damage from the pathogens (fungi, bacteria and viruses), which reduce crop yield due to imbalances in plant hormones and nutritional status as well as physiological disorders [2]. In addition, a high loss in global crop production (up to 50%) due to abiotic stresses has been reported [3]. Different abiotic stresses and their impacts on cucumber production have been discussed in many reports, including drought [4], salinity [5], cold stress [6], heavy metals [7] and heat stress [8]. Several investigations have studied the single stress and many combined stresses [1,2,6], whereas global environmental changes refer to these stresses in various combinations [3,9]. A distinguished adaption of the cultivated plants has been noticed under environmental stresses by modulating their secondary metabolites, and gradually developing their defense system [10]. The plant tolerance against multiple stresses could be achieved through exogenous and endogenous signaling molecules, including nitric oxide and hydrogen sulfide [11], hydrogen peroxide [1,7], salicylic acid [7], osmotin [12] and strigolactone [13], as well as some nutrients such as sulfur [14], silicon [15], selenium [16] and paclobutrazol [17]. 

Cucumber (*Cucumis sativus* L.), as a vegetable-horticultural crop, has a great ability to grow under greenhouse conditions all over the world [18]. Cucumber is vulnerable to cold stress and sensitive to heat stress, although it is a thermophilic vegetable [19]. The production of cucumber under greenhouse conditions faces many abiotic/biotic stresses particularly under arid and/or semi-arid conditions because of high applied mineral fertilizers, the salinization of soils and higher temperatures in summer [20]. Several approaches have been applied to overcome these stresses, including grafting. Grafting in cucumber is a common practice by which resistant/tolerant plants for stresses could be performed using resistant/tolerant rootstock against biotic stress such as fusarium wilt [21] and abiotic stress [22] such as heat stress [8]. 

Many agricultural practices have been changed due to several environmental stresses, including the agronomic, breeding and genetic programs. Grafting is an important agronomic technique that could save the costs and time of breeding programs [2]. Grafted vegetables onto tolerant or resistant rootstocks can cope with different purposes, including managing soilborne diseases such as *Fusarium oxysporum* [23], increasing plant tolerance to abiotic stress such as heat stress [8], increasing the vigor of plants and enhancing the water and nutrients uptake [24]. Grafted plants are common for vegetable crops in greenhouse cultivation, in particular, the cucurbitaceous against fusarium wilt and the solanaceous against bacterial wilt [21]. The most common vegetable plants, which are grafted before transplanting into the field or greenhouse, include cantaloupe [25], cucumber [2], tomato [5], pepper [26] and watermelon [27]. The most important benefit of grafting may be represented in decreasing the consumed agro-chemicals against soilborne pathogens, which provides an environment friendly technique and integrated crop manager [28]. 

Plant pathogens are considered one of the most important biotic stresses, which cause a significant loss in cucumber crop production (up to 60%), in particular, under greenhouse conditions [29]. Greenhouse cucumber has many problematic diseases limiting the crop production such as powdery mildew (*Podosphaera xanthii*), pythium crown (*Pythium aphanidermatum*), fusarium wilt (*F. oxysporum*), root-knot nematode (e.g., *Meloidogyne incognita*), gummy stem blight (*Didymella bryoniae*) [30] and cucumber green mottle mosaic virus [31]. The management of these pathogens could be achieved through many approaches such as chemical control using fungicides, the biological control, solarization and bio-fumigation, grafting [32] and nanoparticles [33]. There are many types of pathogens that cause wilt diseases in horticultural plants including viral wilt [34], bacterial wilt (e.g., *Ralstonia solanacearum, Erwinia tracheiphila*) [35,36,37], fungal wilt (*Fusarium* spp. and *Verticillium* sp.) [38,39], and root-knot nematodes (*Meloidogyne incognita*, *M. javanica*, *M. arenaria*, and *M. hapla*) [28]. These pathogens may cause damage up to 100% of losses in crop productivity as reported for bacterial wilt [28]. The wilt symptoms start from tyloses production in the xylem vessels as an indicator of infection by vascular wilt pathogens, followed by blocks in the vessels [40]. The fusarium wilt is considered one of the major pathogens (i.e., soil-borne fungal diseases), which constrains cucumber yield and its quality [29] and occurs at any cucumber growth stage [41]. Cucumber fusarium wilt could be managed through the application of *Trichoderma spp*. [40,42], *Bacillus amyloliquefaciens* [43], green manures [44] and grafting onto different resistant rootstocks [45]. However, fusarium risk is increased by high or low temperature stress, high spore load and poor soil drainage. 

Therefore, the objective of this work was to assess the response of grafted cucumber onto five cucurbit rootstocks grown in pots to heat stress singly or combined with fusarium wilt under greenhouse conditions. In addition, the biochemical and anatomical parameters were investigated under studied conditions.

## 2. Results

### 2.1. Evaluation of Grafted and No-Grafted Cucumber Plants against Fusarium Wilt under Heat Stress

Cucumber hybrid “Gianco F_1_” was selected as a scion in this study. The scion was grafted onto five different cucurbit rootstocks (i.e., Bottle gourd, Cobalt, Ferro, VSS-61 F1 and Super Shintoza), and evaluated for resistance to fusarium wilt singly or combined with heat stress conditions at the experimental locations. A comparison among these cucurbit rootstocks and their evaluation based on disease incidence (%), wilt reduction (%), disease severity (%) and efficiency percent were significantly differed in the two locations as shown in Table 1 and Figure 1. The lowest disease incidence of fusarium wilt (20.8 and 16.6%) was recorded for cultivated plants of cucumber grafted onto the rootstock of VSS-61 F_1_ and the highest wilt reduction (79.2 and 83.4%, respectively in both experimental locations) compared to non-grafted cucumber plants which showed the highest disease incidence of fusarium wilt. Concerning the efficiency percent, the rootstock VSS-61 F_1_ had the highest efficiency (85 and 88%, respectively in both experiments) and recorded the lowest disease severity percent of fusarium wilt (15.3 and 12%, respectively in both locations) compared with the cucumber plants grafted onto other cucurbit rootstocks and control plants (non-grafted). These results might confirm that VSS-61 F_1_ was the best cucurbit rootstock for the cucumber hybrid “Gianco F_1_” grafted against fusarium wilt under heat stress conditions. It could also be that the disease incidence percent was increased by increasing the period of infection with fusarium after transplanting up to 75 days. In contrast, the rootstock “Bottle gourd” was not suitable to be a rootstock for cucumber grafting under studied conditions because of its lowest grafting efficiency (25 and 21%, in both locations, respectively), its high fusarium incidence (79.1 and 75.4%) and disease severity of fusarium wilt (76 and 79%, respectively in both locations) compared to the other rootstocks. 

### 2.2. Anatomical Structure of Grafted Cucumber under Disease Pressure and Heat Stress

The most effective mechanism of rootstock for resistance to fusarium wilt disease might include producing antifungal compounds, forming papillae at sites of penetration, the suberization and lignification of cell walls, the accumulation of gums, gels or tyloses and crushing within xylem cells and vessels by the proliferation of adjacent parenchymal cells [46]. The concrescence characteristics of the grafted cucumber as a scion union with rootstocks of various compatibilities are shown in Figure 2. The compatibility between the rootstock VSS-61 F_1_ and the scion was found to be quickly developed as shown in Figure 2 (photo 2), whereas Bottle gourd rootstock did not exhibit this behavior due to forming a necrotic layer as shown in Figure 2 (photo 1). Based on this result, the pathogen progression was effectively prevented by resistant rootstock VSS-61 F_1_ through the lignification of the cell walls and the accumulation of polyphenolic compounds and carbohydrates in xylem vessels of the rootstock. This increased the limiting fungal growth in the outer stem area, and it was primarily associated with lignification of host cell walls, and detected by their red coloration. The thickness of lignin was constitutively accumulated on the outer cell walls of the epidermis, exodermis, cortex and vascular stele. These were strongly associated with the resistance levels measured for rootstock samples.

The detection of different compounds within xylem vessels and intercellular spaces surrounding it as well as the lignification of cell walls are shown in Figure 2 (photos 3, 4 and 5). Staining with AGS showed that the key substances after this staining were mucilage, a polysaccharide, demonstrated by its dark green coloration in xylem vessels of rootstock VSS-61 F_1_, as shown in Figure 2 (photo 9). The accumulation of polyphenols inside the xylem vessels of rootstock VSS-61 F_1_ was shown by its dark red coloring after AGS staining, as shown in Figure 2 (photo 7). Polyphenol accumulation was more frequent in the resistant rootstock VSS-61 F_1_, where some proto-xylem cells were completely clogged with polyphenols. Bottle gourd was more susceptible where polyphenols and carbohydrate accumulation were not observed in its xylem vessels, as shown in Figure 2 (photos 6 and 8). The accumulation was, however, only clearly noticed in the resistant rootstock VSS-61 F_1_.

### 2.3. Vegetative Growth in Non-Infected Grafted and Non-Grafted Plants under Heat Stress

The stem length and stem diameter for both scion and rootstocks were measured as vegetative growth parameters. It is obvious generally that the plant stem length and stem diameter for scion and rootstocks were significantly increased at 70 DAT compared to the control plants in both locations (Table 2). Under abiotic stress, the stem diameter (for both scion and rootstocks) and stem length of non-grafted plants (control) were recorded at the lowest values (9.38 and 9.4 mm; as well as 217.25 and 222.3 cm, respectively in both locations) in general for all studied rootstocks, whereas both grafted rootstocks Ferro and VSS-61 F_1_ recorded the highest values (12.54 and 12.45 mm in the first location; 12.64 and 12.55 mm in the second location for rootstock; 10.84 and 11.13 mm in the first location; 10.2 and 10.8 mm in the second location for scion at 30 DAT). In the same line, both of VSS-61 F_1_ and Super Shintoza showed the tallest stem length (295.16 and 293.65 cm in the first location; 304.05 and 299.95 cm in the second location, respectively) at 70 DAT. 

The impact of grafting onto different rootstocks for both (Fv/Fm) and (Fv/Fo) in cucumber leaves is presented in Table 2. The highest significant values for Fv/Fm and Fv/Fo were recorded by VSS-61 F_1_ rootstock (0.83 and 2.55 in the first location; 0.9 and 2.67 in the second location, respectively), followed by Ferro rootstock compared to control plants (non-grafted), which resulted in the lowest values for the photosynthetic rate (0.61 and 2.07 in the first location; 0.6 and 2.02 in the second location, respectively). 

### 2.4. Fruit Yields and Its Quality in Non-Infected Grafted and Non-Grafted Plants under Heat Stress

Total yield of cucumber was evaluated by measuring the weight of fruits per plant and their quality (fruit length, fruit diameter, fruit dry matter and ascorbic acid content) in both experimental locations as presented in Figure 3 and Figure 4. There were significant effects due to different rootstocks on fruit yield. The results showed that the highest values of fruit weights were significantly produced by cucumber grafted onto VSS-61 F_1_ rootstock followed by both Ferro and Super Shintoza rootstocks without insignificant differences in between compared with the control plants in both locations. The lowest fruit yield was recorded by non-grafted plants (control).

Data in Figure 4 show that fruit length and ascorbic acid content of cucumber fruits were significantly influenced by grafting onto different rootstocks in comparison with the control (non-grafted plants) only in the second location. Grafted cucumber with all studied rootstocks significantly increased the dry matter content (%) compared to non-grafted plants in both experimental locations. The highest dry matter percent (4.16 and 4.19%, respectively in both locations) was a result of plants grafted on VSS- 61 F_1_ rootstock, whereas the lowest dry matter percent (3.64 and 3.68%, respectively in both locations) was presented by cucumber plants grafted onto Bottle gourd rootstock in most cases. Fruit diameter of cucumber was not significantly influenced by grafting onto different rootstocks in both locations.

## 3. Discussion

The establishment of a million greenhouses through the national Egyptian project has been focused particularly on vegetable and fruit production, which will contribute to the “sustainable development” strategy for 2030. The cucumber production under greenhouses in arid environments, especially in developing countries, in summer represents a real challenge. Most of these greenhouses in Egypt are considered “low-cost structures”. These greenhouses have fewer facilities that create many stressful conditions in greenhouse cucumber production, including biotic and abiotic stresses [47]. The main problems facing the greenhouse cucumber production may include soil salinization and degradation, nitrate groundwater pollution and heat stress [48] as well as plant pathogens such as bacteria, fungi, viruses and nematodes [30]. These phyto-pathogens and diseases may cause damage and a decline in the yield of greenhouse cucumber such as fusarium wilt (*F. oxysporum* f. sp. cucumerinum), powdery mildew (*Podosphaera xanthii*), gummy stem blight (*Didymella bryoniae*) and Alternaria blight [30]. These problems have also become major obstacles in greenhouse cucumber production under changing climate, particularly high-temperature stress [49].

The grafting technique has been developed over the last century for cultivation of the vegetables and has been adapted to adverse environments. This technique has a distinguished relationship between the rootstock, which should be tolerant or resistant to stresses and its root system has the ability to improve the uptake of water and nutrients from scion making the grafted plant stronger compared to non-grafted plants. The effective uptake of water and nutrients by grafted plants may support and promote their adaptability to stressful environments and lowering the required agrochemicals (e.g., mineral fertilizers and pesticides) during plant cultivation [50]. The grafting has been applied as a tool to improve the fruit yield and its quality of cucumber under salt stress [51] and water use efficiency under drought [52]. The grafting also may support the resistance of greenhouse cucumber against the wilt disease that caused by the fungal pathogen (*F. oxysporum* f. sp. cucumerinum), as a main destructive soil-borne disease infecting cucumber [23]. The management of greenhouse cucumber wilt includes the application of biocontrol agents such as *Paenibacillus polymyxa*, *Trichoderma* sp., mycorrhiza and *Streptomyces* sp. [53], as well as chemical control agents such as silicate, and other practices such as fumigation, crop rotation [44] and grafting [54].

In this study, the productivity of greenhouse cucumber was investigated by grafting onto five cucurbit rootstocks under fusarium wilt as biotic stress and heat stress. The grafting of greenhouse cucumber also was handled as a promising tool in dealing with the resistance to fusarium wilt under high-temperature stress. No previous study has investigated the multiple stresses, particularly for combined fusarium wilt and heat stress on the production of grafted greenhouse cucumber, whereas a published article by Bayoumi et al. included combined abiotic stresses (i.e., salinity and heat stress) [2]. Recently, the production of grafted cucumber under a single stress has been reported, such as salinity stress [45,55], water stress [56], fusarium wilt [39] or under chilling stress [57]. The splice grafting technique has been found to alleviate stem rot caused by *Fusarium equiseti* and *F. proliferatum* through promoting the histological and biochemical (antioxidant enzymes and total phenols) defense response of grafted seedlings [39]. 

Five different rootstocks of cucurbits (i.e., Super Shintoza, Bottle gourd, VSS-61 F_1_, Cobalt and Ferro) were selected in this study. These rootstocks are commonly used by Egyptian farmers for grafting cucumber. A limited number of studies have included these rootstocks for grafting cucumber under Egyptian conditions, in particular, under heat stress [58], and for grafting watermelon under cold stress [59]. The grafting of cucumber hybrid onto five cucurbit rootstocks has been evaluated to determine which rootstock was resistant to fusarium wilt under high-temperature stress under greenhouse conditions. The VSS-61 F_1_ rootstock was the best among the studied rootstocks for grafting cucumber. This rootstock had a high grafting efficiency, which resulted in a high compatibility between the rootstock and the scion, and provided high levels of resistance against to fusarium wilt under heat stress. The VSS-61 F_1_ rootstock also has a vigorous root system, including many lateral roots and root hairs, large main roots and total root length and root surface area [60]. The anatomical section also might confirm that anatomical and histo-chemical findings combined with spectral examination of the epidermis, exodermis, cortex and vascular stele effectively hinder development of the fungi. Similar results have been obtained by Sabry et al. [39], who confirmed that grafted cucumber can form some compounds as protective substances such as necrotic layer to prevent fusarium invasion on the histological level. The main components of these different barriers were carbohydrates and the role of lignin deposition in fusarium resistance, indicated by phenolic compounds such as lignin. The thickness of lignin constitutively accumulated on the outer cell walls of the epidermis, exodermis, cortex and vascular stele. These were strongly associated with the resistance level measured for rootstocks. The accumulation of phenolic precursors in lignin was observed in the resistant plants inoculated with fusarium but not in the susceptible one. Therefore, the grafted cucumber plants can ameliorate the severity caused by *Fusarium* spp., such as *F. oxysporum* f. sp. *cucumerinum* in the present study, and *F. equiseti* and *F. proliferatum* as reported by Sabry et al. [39]. The behavior in each rootstock may be controlled by a kind of stress (biotic or abiotic stress) and a horticultural crop species as explained earlier. 

The activity of *Fusarium oxysporum* might be affected by the temperature, and the optimum temperature ranges from 25 to 28 °C [61]. High temperatures have been reported to influence the fusarium development and aggressiveness. This fungus could reduce the production of greenhouse cucumber up to 100%, causing a serious economic loss in the yield and its quality, especially if the infection happens at the early growth stages [53]. The main symptoms of cucumber fusarium wilt may be resulted from the fungal toxins and/or blocking the water transport, which might link with the injury of leaf cell membrane and damage the cells due to non-control of the water loss from leaves [62]. The fusarium wilt and its severity may be accelerated with the high-temperature stress, for which it may also decrease the yield and quality of cucumber, especially the production during the summer in greenhouses [63,64,65]. On the biochemical level, grafted cucumber plants show an increase in the level of antioxidant enzymes and total phenols under fusarium stress compared to non-grated plants, as reported by Sabry et al. [39]. 

Therefore, grafting cucumber is a promising technique for stressful cultivated plants under biotic and/or abiotic stresses (Figure 5). Concerning the suggested mechanism for both abiotic and biotic stress, it might be differ depending on kind of stress, pathogen type and cultivated plant species (Figure 5). The suggested mechanism may include forming higher levels of antioxidant enzymes activities and total phenolic content compared to non-grafted ones [39], reducing horizontal and vertical movement of pathogens within the xylem, reducing its multiplication within the infected tissues and suberization and lignification of cell walls [40], as well as regulate the ABA-dependent H_2_O_2_-driven mechanism [57]. 

Under heat stress conditions, cucumber leaves reduce and the seedlings will be thin due to the inhibition of different plant biological processes such as metabolism and photosynthesis, especially the biosynthesis of proteins and others. Therefore, it could improve the adaptability of greenhouse cucumber to the stress of high temperatures under fusarium wilt depending on the selected rootstock for cucumber grafting and its characterization. High temperatures might alter plant responses to pathogens, where it is important to examine plant responses to stress combinations under different scenarios [66]. The relationship between high temperature and fusarium wilt incidence has been studied in many horticultural crops, such as cabbage [67], watermelon [68] and tomato [69,70]. These studies concluded that the incidence in fusarium wilt is increasing with rising global temperature. However, the interaction between heat stress and *Fusarium* resistance on cucumber remains unknown. In the present study, heat stress increased disease incidence in cucumber indicating that genetic improvement of cucurbit rootstocks for grafting cucumber under heat stress is urgently needed. So, high-temperature stress may turn the non-host rootstocks into host plants and fail to protect the grafted cucumber plants. Based on the obtained results, Bottle gourd became susceptible to fusarium wilt, resistance levels in Super Shintoza Cobalt and Ferro decreased under heat stress, but VSS-61 F_1_ remained resistant. Therefore, VSS-61 F_1_ is the most recommended rootstock for grafting cucumber under heat stress in Egypt. Further studies are needed to test disease-resistant rootstocks under different abiotic stress, singly or in combination, to select rootstocks that will best suit climate changes.

It is worth mentioning that there are many published studies on the role of grafting under stress, including plant disease resistance and heat tolerance on different horticultural crops along with cucumber. Balfagón et al. [71] studied the role of grafting in improving the tolerance of citrus to combined heat and drought stresses by enhancing the metabolic (e.g., changes in carbohydrate and amino acid fluxes) and hormonal (i.e., abscisic, indoleacetic, salicylic and jasmonic acids) response to stress. Grafting in *Hylocereus* also can support biochemical, physiological and molecular parameters of the grafted plants under extreme high temperatures (45/35 °C day/night) [72]. Concerning the stress of plant disease, many studies have reported the effective role of grafting for improving cultivated plants to resistance, such as the late blight in potato [73] or *Phytophthora capsica* in pepper [74]. Apart from the stress of plant disease and heat, grafting is also considered an effective strategy to reduce cadmium (Cd) accumulation in the aerial plant parts in grafted eggplant grown in polluted soil with Cd [75,76,77,78]. 

## 4. Materials and Methods

The experiments were carried out at two different locations in Kafr El-Sheikh governorate during the late summer season in 2021 (from May to August), which causes heat stress in Egypt at the first location “Kafr El-Sheikh” (the protected cultivation center, Horticulture Department, Faculty of Agriculture, Kafrelsheikh University), and “Sidi Salem”, a city located at Kafr El-Sheikh governorate, Egypt (the second location). The details of the experiment including studied stress, growth media, experimental duration and locations, grafting methods and measured parameters are described in Table 3. The vegetative growth, fruit yield and quality of grafted cucumber onto each cucurbit rootstock also were investigated under studied abiotic stress in the two locations.

### 4.1. Selected Cucumber Scion and Cucurbit Rootstocks

Cucumber hybrid “Gianco F1” was utilized as a common hybrid under high-temperature conditions. It is susceptible to *F. oxysporum* f. sp. cucumerinum. Seeds were obtained from Rijk Zwaan Company in Egypt (Agro Company, Cairo, Egypt). Five different rootstocks for cucurbits (i.e., Super Shintoza, Bottle gourd, VSS-61 F1, Cobalt and Ferro) were chosen in this work (Table 4), whereas the control was non-grafted cucumber. Seeds of cucumber hybrid and different rootstocks were cultivated in seedling Styrofoam trays (84 compartments) that were filled with the common media of both coco peat and vermiculite (1:1; *v*/*v*) under greenhouse in the middle of April 2021 for the two locations. The different steps for grafting are presented in Figure 6. The grafting was conducted using the slant cut grafting method after 4 weeks from the sowing date of both cucumber scion and five cucurbit rootstocks. All cucurbit rootstocks under study are commonly used by farmers in Egypt and this study aimed to confirm their resistance to this disease under heat stress conditions to determine the degree of resistance or the resistance behavior with the hybrid cucumber used as a scion under high-temperature conditions.

### 4.2. Growth Conditions

Successful grafted seedlings of cucumber on different five cucurbit rootstocks (6 weeks old) and control (non-grafted plants) were transplanted in pots under greenhouse conditions. The pots for the experiments were divided into two groups: the experiment for the first group was performed to investigate the response of grafted cucumber seedlings grown under greenhouse to Fusarium wilt disease under high-temperature stress in the late summer season, and the experiment for the second group was conducted without Fusarium infection to investigate the effect of grafting onto rootstocks on growth, yield and fruit quality of cucumber under heat stress conditions. The mean daily temperature and relative humidity (at 2 m) under the greenhouse during the growth season of cucumber reached between 26.1 and 43.0 °C (night and day) and 43.33–86.33% for the first location and 24.0 and 40.2 °C (night and day) and 42.5–88.67% for the second location, respectively. The maximum and minimum temperatures in the first location were 22.3 and 47.3 °C, and in the second location were 20.3 and 43.5 °C, respectively. The mean values for maximum and minimum air temperature are presented in Figure 7. The successful grafted cucumber plants were irrigated every day and supplied with fertilizers twice a week with suggested doses of nutrient solution, starting from the first week after transplanting. 

#### 4.2.1. Isolation and Identification of the Fungus

Cucumber plants, which showed wilt disease symptoms, were collected from Kafr El-Sheikh Governorate. Roots of the diseased plants were washed by water to clear any involving soil parts. Small pieces of the diseased tissues were disinfected with 3% solution of sodium hypo-chloride for 2–3 min. Thereafter, they were dried using sterilized filter paper and transported into petri dishes containing water agar medium. Dishes were incubated at 27 °C from 24 to 48 hr. Hyphal tip cultures of grown fungi were maintained on potato dextrose agar. Fusarium isolate was prepared in single spore cultures and was identified according to [79]. The fungus was confirmed as *Fusarium oxysporum* f. sp. cucumerinum by sequencing of the ITS regions with reference to the GenBank number KT461496 by Kamel, S.M. Professor of Vegetable Diseases, Plant Pathol. Res. Inst., ARC, Giza, Egypt.

#### 4.2.2. Inoculation Test

Artificial inoculation was performed on 40 grafted seedlings (per each rootstock), plastic pots (diameter of 30 cm) were sanitized by immersing in formalin solution (5%) for 15 min and then put in open air to formalin evaporation. Grafted cucumber seedlings were transplanted in pots containing sterilized sandy loam soil artificially infected with *Fusarium oxysporum* f. sp. Cucumerinum; the pathogen causes cucumber wilt, which was previously grown on sand barley medium in 500 mL bottles for 15 days at the rate 5% of the soil weight for each pot. Pots were watered every 2 days for 2 weeks to ensure the establishment of the Fusarium isolate in the soil. Both experiments in studied locations were coordinated in the design completely randomized with five replications; each replicate had four pots for each rootstock and control treatment (non-grafted cucumber) under greenhouse conditions. Each pot was transplanted with two grafted cucumber seedlings.

#### 4.2.3. Recording Wilt Severity

The wilt disease was evaluated through recording and calculating the following items: disease incidence (DI), disease reduction, disease severity (DS) and efficiency percent (E). The disease incidence percent was recorded periodically at 15 days interval (from 30 to 75 days) using the formulation: 

Disease incidence percentage = (number of wilted plants)/(total number of plants) × 100

Disease reduction percent was recorded after 75 days using formula:
Disease reduction (%) = (A − B/A) × 100
where: A and B represent disease incidence percent of control and treatment, respectively.

Disease severity (%) on individual plants was calculated on a scale from 0 to 5, as follows: 0, 1, 2, 3, 4 and 5 to represent no symptoms, <25, 25–50, 50–75, 76–100% of leaves with symptoms and completely dead plant, respectively according to [80].

Disease severity percent (DS) was determined according to equation:

DS (%) = [Σ (rating no.) (no. plants in rating category)/(Total no. plants) (highest rating value)] × 100

The efficiency percent (E) was recorded after 75 days using the following formula:

E (%) = (DS as% in control − DS as% in treatment)/DS as% in control × 100

### 4.3. Anatomical Structure

For the anatomical investigation and the interpretation of the mechanism for resistance to disease or sensitivity, transverse sections were obtained from the grafted union region 70 day after transplanting (DAT). The samples were set for 48 h in FAA solution (formaldehyde + alcohol ethanol + glacial Acetic acid 5, 50, 10%, respectively and 35% water) and then processed in 70% alcohol before use. Parts were dehydrated (75–100%) in the alcohol series and coated with warm paraffin (57 °C) [81]. The resulted blocks were then cut using rotary microtome in 10 μm sections and stained with Alcian Green Safranin (AGS). Using this discoloration, carbohydrates (including mucilage and cell walls) were colored green, blue or yellow, while the walls were cutinized, lignified and suberized, but lipid and tannin materials inside cells seemed red [82]. Slices were detected by a transmission microscope (Leica Microsystems, Wetzlar, Germany) and a digital camera for photographing (Nikon DXM1200F, The Netherlands).

### 4.4. Vegetative Growth, Yield and Its Quality

Main stem diameter (mm) for both rootstock and scion and stem length (cm) as growth parameters were determined at 30 and 70 DAT. These were recorded on five random plants for each treatment (20 plants in each treatment in both locations). Main diameters were measured under/above unit grafting (2 cm) using a digital caliper. The parameters of chlorophyll fluorescence were recorded with a hand fluorometer (Opti-Sciences, NH, USA). Leaves were acclimatized to darkness with clips for about 20 minutes before recording data. The maximum efficiency of the photosystem (PSII) (Fv/Fm) and maximum primary yield of PSII photochemistry (Fv/Fo) were determined after the dark period [83]. Next, cucumber fruits were harvested at acceptable market maturation stage in the middle of June to estimate the total fruit crop per plant, which continued for about two months. The fruit quality was analyzed using dry matter (%) and ascorbic acid content at 70 DAT. Both experiments were coordinated in a randomized design with five replications, each one containing the five rootstocks and control treatment (non-grafted cucumber) for each group under greenhouse conditions.

### 4.5. Statistical Analyses

Data resulting from the two locations during the 2021 year of study were arranged and statistically analyzed according to [84]. Analysis of variance as well as Duncan’s multiple range tests were carried out by the CO-STAT computer software program.

## 5. Conclusions

Due to the changing climate and global warming, tremendous economic losses are projected for agriculture and all life fields. Heat stress is subjected to decrease the productivity of crops due to the reduction of all yield attributes in addition to plant growth, including all biological processes. Grafting is a promising and sustainable solution against biotic and abiotic stresses for greenhouse cucumber. The investigation of heat stress in combination with other biotic stress (*Fusarium* wilt) was carried out in this study. The current study highlights the importance of heat stress and its biotic stress on grafted greenhouse cucumber, including the adverse impacts on physiological, morphological, biochemical and anatomical responses. The main aim of the experiment was to determine the best rootstock for greenhouse cucumber that could be grafted to fight *Fusarium* wilt disease under high temperatures. It could be recommended that the VSS-61 F1 rootstock is more suitable to be grafted by greenhouse cucumber hybrid under arid environments or studied conditions. The long-term cultivation of greenhouse cucumber in these regions requires further evaluation from all points of views. This knowledge may help to manage the risk of Fusarium in protected cultivation through better selection of resistant rootstock for cucumber production under heat stress, but growing cucumbers under different growing soil media and planting dates may show different reactions.

## Figures and Tables

**Figure 1 plants-11-01147-f001:**
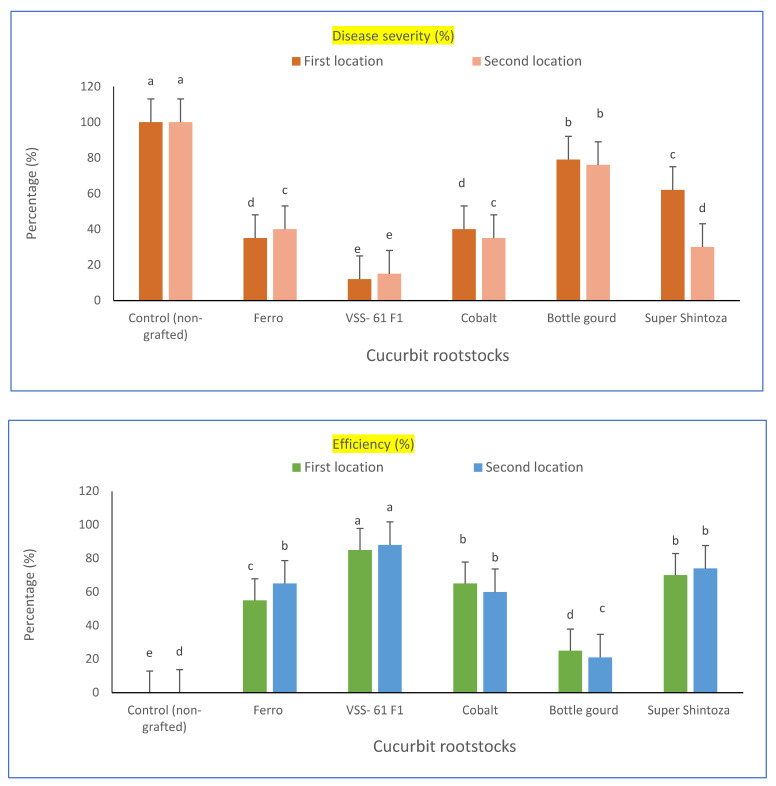
Fusarium wilt disease severity percentage and efficiency percent of greenhouse cucumber plants grafted on different rootstocks under disease pressure and heat stress in both locations. Values are means ± standard deviation (SD) from three replicates. Different letters in the same column show significant differences among each group of treatments according to Duncan’s test at *p* ≤ 0.05.

**Figure 2 plants-11-01147-f002:**
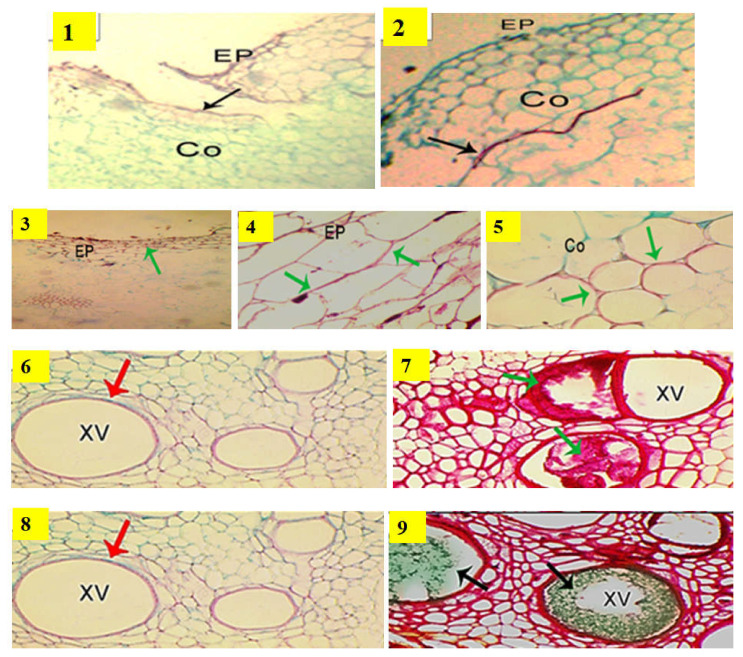
Characteristics of the concrescence of a cucumber grafted union with rootstocks of different compatibilities under disease pressure and heat stress in the most susceptible rootstock, Bottle gourd and the most resistant rootstock, VSS-61 F_1_. Photos 1 and 2 represent the compatibility of grafted union between the rootstock Bottle gourd and the scion; VSS-61 F_1_ rootstock and the scion; respectively. Photos 3 and 4 show cell wall lignification (green arrows) at the epidermis (Ep). Photo 5 shows formation of an exodermis (Ex) to prevent penetration of fusarium hypha as well as the lignification of cortical cell walls (Co) (green arrows), which prevent hypha penetration into cortical cells in stem vascular tissues of resistant rootstock VSS-61 F_1_. Photo 6 shows that accumulation of polyphenol compounds was not detected in xylem vessels (xv) of susceptible rootstock, Bottle gourd; whereas Photo 7 represents accumulation of polyphenols (red staining indicated by green arrowheads) in xylem vessels of resistant rootstock VSS-61 F_1_. Photo 8 shows that accumulation of carbohydrates was not detected in xylem vessels of susceptible rootstock, Bottle gourd; whereas Photo 9 represents accumulation of carbohydrates (green staining indicated by black arrowheads) in xylem vessels of resistant rootstock VSS-61 F_1_. Control treatment (non-grafted plant) is not shown here because all control plants died due to the infection by *Fusarium* wilt.

**Figure 3 plants-11-01147-f003:**
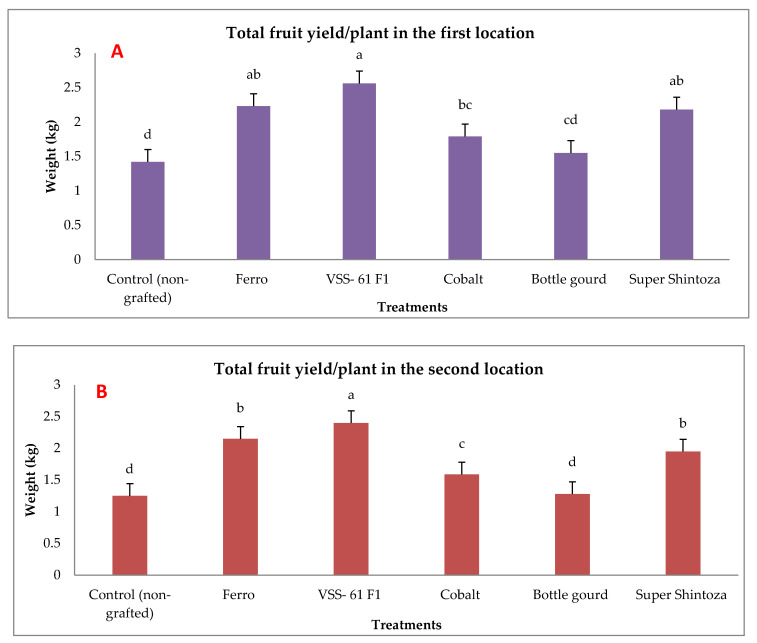
Effects of grafting onto different rootstocks on total fruit yield of cucumber (kg plant^−1^) under heat stress conditions in the first (**A**) and second location (**B**). Values are means ± standard deviation (SD) from three replicates. Different letters in the same column show significant differences among each group of treatments according to Duncan’s test at *p* ≤ 0.05.

**Figure 4 plants-11-01147-f004:**
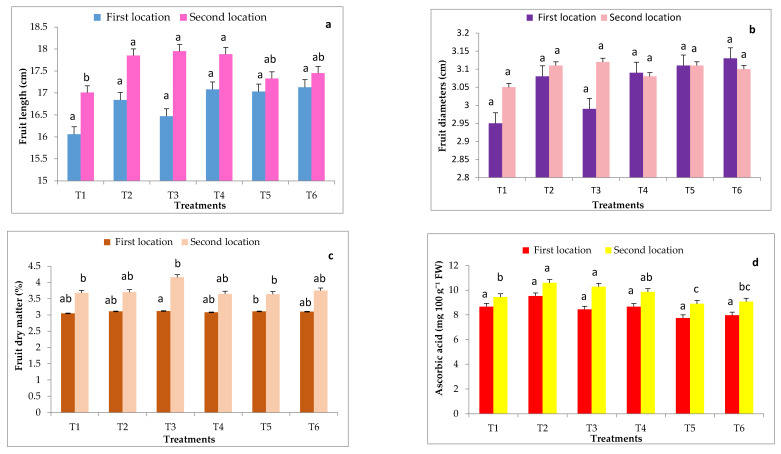
Effects of grafting onto different rootstocks on some fruit quality of cucumber fruits after 70 DAT under heat stress conditions during in both experimental locations including fruit length (**a**), fruit diameter (**b**), fruit dry weight (**c**), ascorbic acid (**d**). Abbreviations: T1, T2, T3, T4, T5 and T6 represent control, Ferro, VSS- 61 F1, Cobalt, Bottle gourd and Super Shintoza. Values are means ± standard deviation (SD) from three replicates. Columns with the same letter are not significant according to Duncan’s multiple range test at *p* ≤ 0.05.

**Figure 5 plants-11-01147-f005:**
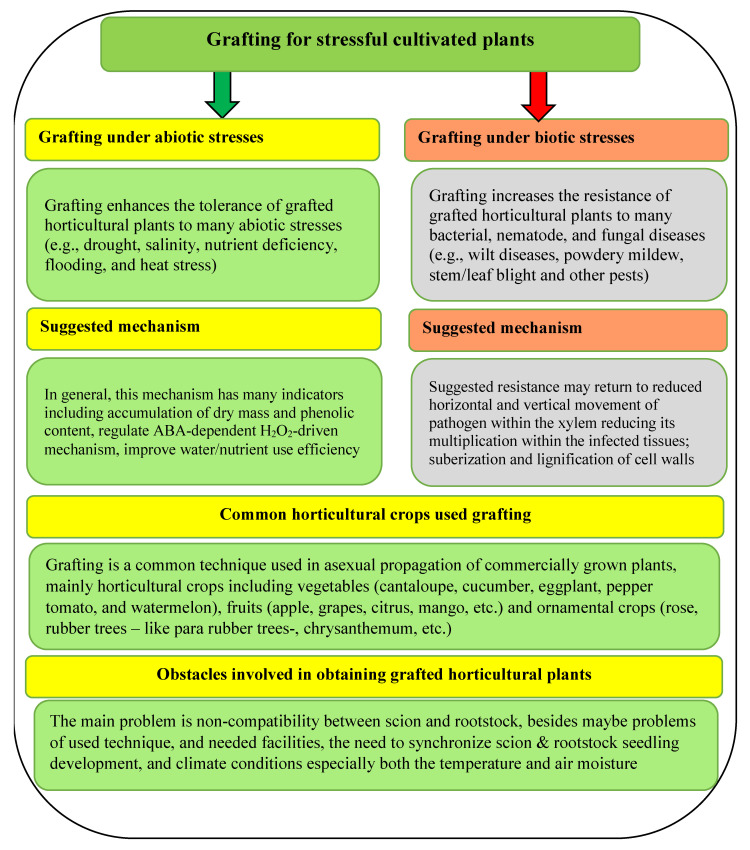
General overview for the grafting process, including the mechanism under biotic and abiotic stress, different plant species using grafting, and the challenges that face this technique. Sources: [28,35,36,37,57,65].

**Figure 6 plants-11-01147-f006:**
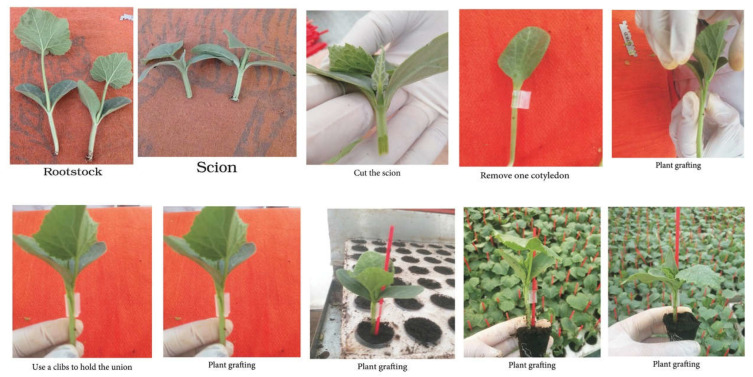
Slant cut grafting method used in grafting for the current study, including the following steps: prepare both the rootstock and scion, then cut the scion, remove one cotyledon of rootstock, match the cut surfaces on the scion and rootstock, use a clip to hold the union and then produce the grafted seedlings (photos by authors).

**Figure 7 plants-11-01147-f007:**
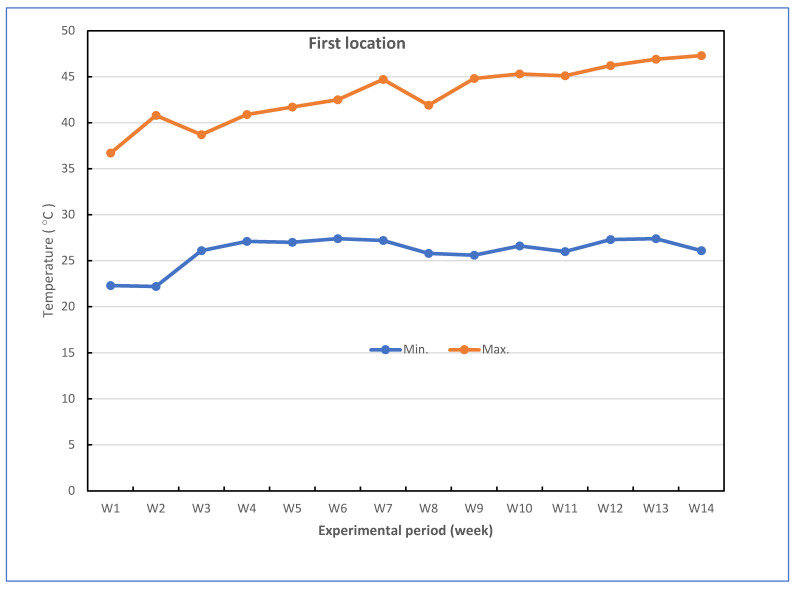
Weekly air temperature inside the net-house conditions during the experiment period in both locations.

**Table 1 plants-11-01147-t001:** Response of cucumber plants grown in greenhouse grafted onto different rootstocks to fusarium wilt under disease pressure and heat stress in both locations.

Cucurbit Rootstocks	Disease Incidence (%)			
	Days after Transplanting			
	30	45	60	75
**First location**				
Control (non-grafted)	50.0 a	83.3 a	100 a	100 a
Ferro	16.6 c	20.5 de	30.3 de	35.0 c
VSS-61 F_1_	8.3 d	16.6 e	20.8 f	20.8 d
Cobalt	16.6 c	33.3 c	41.6 c	41.6 c
Bottle gourd	33.3 b	50.0 b	66.6 b	79.1 b
Super Shintoza	8.3 d	20.8 de	25.0 ef	33.3 c
F-test	*	**	**	**
**Second location**				
Control (non-grafted)	40.0 a	65.3 a	85.5 a	100 a
Ferro	12.4 c	15.5 de	25.6 d	31.5 c
VSS-61 F_1_	5.0 d	10.8 e	16.6 e	16.6 d
Cobalt	12.6 c	24.6 c	30.3 c	35.9 c
Bottle gourd	30.6 b	50.0 b	66.6 b	75.4 b
Super Shintoza	5.3 d	15.6 de	25.0 d	31.2 c
F-test	*	**	**	**

The signs ** and * indicate significant differences at *p* < 0.01 and *p* < 0.05, respectively, according to F. test, Different letters in the same column show significant differences among each group of treatments according to Duncan’s test at *p* ≤ 0.05. Number of samples was 20 plants as 4 plants per each replicate (4 plants × 5 replicates).

**Table 2 plants-11-01147-t002:** Response of main stem diameter, stem length and chlorophyll fluorescence to grafting on different rootstocks after 30 and 70 DAT to heat stress in both experimental locations.

Treatments	After 30-Day Transplanting		After 70-Day Transplanting		
	Main Stem Diameter (mm)		Stem Length (cm)	Chlorophyll Fluorescence	
	Rootstock	Scion		FV/FM	FV/F0
**First location**					
Control (non-grafted)	10.17 c	9.38 c	217.25 b	0.61 c	2.07 c
Ferro	12.56 a	10.84 ab	256.87 ab	0.75 ab	2.48 a
VSS—61 F_1_	12.45 a	11.13 a	295.16 a	0.83 a	2.55 a
Cobalt	12.09 ab	9.95 bc	248.18 ab	0.72 ab	2.22 b
Bottle gourd	11.53 b	9.78 c	240.09 ab	0.68 b	2.15 b
Super Shintoza	12.23 ab	9.84 bc	293.65 a	0.74 ab	2.33 ab
F-test	**	**	*	**	**
**Second location**					
Control (non-grafted)	10.40 c	9.40 c	222.30 b	0.60 c	2.02 c
Ferro	12.64 a	10.20 ab	265.80 ab	0.79 ab	2.49 b
VSS—61 F_1_	12.55 a	10.80 a	304.05 a	0.90 a	2.67 a
Cobalt	12.33 a	9.90 bc	254.20 ab	0.77 ab	2.48 b
Bottle gourd	12.02 b	9.80 bc	248.40 ab	0.73 b	2.24 bc
Super Shintoza	12.09 b	9.90 bc	299.95 a	0.78 ab	2.47 b
F-test	*	*	**	**	**

The signs ** and * indicate significant differences at *p* < 0.01 and *p* < 0.05, respectively, according to F. test. Different letters in the same column show significant differences among each group of treatments according to Duncan’s test at *p* ≤ 0.05. Number of samples was 20 plants as 4 plants per each replicate (4 plants × 5 replicates).

**Table 3 plants-11-01147-t003:** Experimentation details for the current study.

Item	The Experiment Details
Studied stress	Combined stress including Fusarium wilt disease (biotic) and heat stress (abiotic)
Growth media	Sandy loam soil infested with and without *F. oxysporum* f. sp. *cucumerinum* infection (the soil pH was 7.80 and soil salinity was 3.32 dS m^−1^ used in the two locations)
Experiment duration	105 days for each location
Experimental location	Two locations (i.e., Kafr El-Sheikh and Sidi Salem)
Grafting details	Cucumber hybrid grafted onto five cucurbit rootstocks
Measured parameters	Fusarium wilt disease parameters and evidence of resistance to Fusarium by anatomical study for grafted plants infested with Fusarium, and growth, yield and fruit quality Parameters for grafted plants without infested Fusarium

**Table 4 plants-11-01147-t004:** Details of five cucurbit rootstocks used in our study.

Commercial and Scientific Name	Place of Seeds	Description
Ferro, *Cucurbita maxima* × *C. moschata*	Rijk Zwaan Company, Cairo, Egypt	Highly resistant to wilt of *Fusarium* and *Verticillium*
Cobalt, *Cucurbita maxima* × *C. moschata*	Rijk Zwaan Company, Cairo, Egypt	Tolerant to high temperature, Proof against *Fusarium* wilt
VSS-61 F_1_, *C. pepo* (Summer squash)	Techno green seed Company, Cairo, Egypt	Proof against both of *Fusarium* wilt and nematode
Bottle gourd, *Lagenaria siceraria* L.	Al-Reda Nurseries, Cairo, Egypt	Proof against *Fusarium* wilt
Super Shintoza, *Cucurbita maxima* × *C. moschata*	Al-Reda Nurseries, Cairo, Egypt	Tolerant to high temperature and impervious to *Fusarium* wilt

## Data Availability

Not applicable.

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
