# Peer review of "Can Grafting Manage Fusarium Wilt Disease of Cucumber and Increase Productivity under Heat Stress?"

_plants, 2022, doi:10.3390/plants11091147_

Round 1
Reviewer 1 Report
Adds more details on differences in response to wilt pathogens
The authors should add other details on any differences in these rootstock combinations to other pathogen
Author Response
Reviewer #1
Comments and Suggestions for Authors
Adds more details on differences in response to wilt pathogens
Response: thanks, we added in the introduction section from lines 93 to 100 in the revised version.
The authors should add other details on any differences in these rootstock combinations to other pathogen
Response: thanks for your comment, In Egypt, fusarium wilt is the main soil borne disease in cucumber especially under heat conditions and farmers apply many chemicals to manage this disease. Therefore, the present study focused on using grafting as applicable technology by farmers to control this pathogen.
We have added more information on using grafting for controlling other soil borne diseases in introduction and discussion sections from lines 286 to 304 in the revised version.

Reviewer 2 Report
Control of pests and diseases in horticulture crops grown under stress conditions, such as heat stress in this paper, is of great interest. Here the authors study the development of Fusarium oxysporum f. sp. cucumerinum (Fusarium wilt) in grafted and non-grafted cucumber during the late season with high temperatures. I understand that the objective is to see if cucumber grafted unto Cucurbita rootstocks protects the plants against the disease under high-temperature conditions.
The temperature conditions of the study are mentioned as daily day and night MEAN temperatures, in point 4,2 Growth conditions. Under greenhouse conditions, temperatures fluctuate, and generally show peak temperatures (when studying heat stresses) which can be very significant. Please present also the maximum and minimum temperatures during the study.
On line 196 Table 4 is cited, when it should be Table 2.
In general, the manuscript is acceptably written but still needs extensive editing of the English language.
Since Fusarium oxysporum f. sp. cucumerinum specifically infects cucumis, and the Cucurbita rootstocks are non-host to the pathogen, the question would be: does high-temperature stress turn the non-host rootstocks into host plants, and then fail to protect the cucumber plants? That would be an important issue to deal with in the discussion, which is missing.
Author Response
Reviewer #2
Comments and Suggestions for Authors
Control of pests and diseases in horticulture crops grown under stress conditions, such as heat stress in this paper, is of great interest. Here the authors study the development of Fusarium oxysporum f. sp. cucumerinum (Fusarium wilt) in grafted and non-grafted cucumber during the late season with high temperatures. I understand that the objective is to see if cucumber grafted unto Cucurbita rootstocks protects the plants against the disease under high-temperature conditions.
The temperature conditions of the study are mentioned as daily day and night MEAN temperatures, in point 4,2 Growth conditions. Under greenhouse conditions, temperatures fluctuate, and generally show peak temperatures (when studying heat stresses) which can be very significant. Please present also the maximum and minimum temperatures during the study.
Response: thanks, we have added the following information in M&M section in the revised MS (Lines 420-422). The maximum and minimum temperatures in the first location were 23.5 and 44.6 °C, and in the second location were 22.1 and 43.4 °C, respectively and figure no. 7 as well.
On line 196 Table 4 is cited, when it should be Table 2.
Response: We agree with your comment. This has corrected in line 196.
In general, the manuscript is acceptably written but still needs extensive editing of the English language.
Response: The MS has been edited for English language.
Since Fusarium oxysporum f. sp. cucumerinum specifically infects cucumis, and the Cucurbita rootstocks are non-host to the pathogen, the question would be: does high-temperature stress turn the non-host rootstocks into host plants, and then fail to protect the cucumber plants? That would be an important issue to deal with in the discussion, which is missing.
Response: Thanks so much for the comment. We agree with your comment. We have added this interesting information in discussion section.
Yes, in this study we used resistant rootstocks, but under heat stress we found that in our work most resistant rootstocks became sensitive to fusarium wilt, So, high-temperature stress turns the non-host rootstocks into host plants and fail to protect the grafted cucumber plants
More Details are mentioned in the revised MS in different places

Reviewer 3 Report
According to the manuscript “Can Grafting Control Fusarium Wilt Disease of Cucumber and its Productivity under Heat Stress?” submitted in the section of plant molecular biology in special issue of control of plant pathogens for a greener future: induced systemic resistance and epigenetics of Plants journal. The authors report the response of grafted cucumber plants onto five cucurbit rootstocks infected with Fusarium oxysporum f. sp. cucumerinum alone and in combination with fusarium wilt disease and heat stress in two different locations in Egypt. The investigation's goal is self-evident. However, the concept is not novel and lack of information on systemic defense mechanisms. It has a critical flaw that must be fixed. Furthermore, the paper's structure must be restructured, and there are a number of issues to be concerned about.
- This research seems a lot like the one you did before “Grafting Improves Fruit Yield of Cucumber Plants Grown under Combined Heat and Soil Salinity Stresses”. To make the work valuable and intriguing to the reader, the author should conduct extra research. It is necessary to investigate the plant's systemic defense mechanisms such as expression of pathogen and stress related genes, defense-related enzymes, and defense-related phytohormones etc.
- Visual data of disease severity and yield size after grafting should be added in order to clearly show the effect between control plants and cucumber grafted plants.
- Table 1 and 2, the authors should explain the meaning of * or ** in the row of F-test and write the number of sample and replicate in each experiment.
- The disease severity and efficiency percentage should divide into 2 figure like figure 1A is disease severity and figure 1B is the efficiency percentage and add the culture locations in same figure.
- Figure 2 is hard to understand since it only shows the structure of VSS-61 F1 and Bottle gourd rootstocks. The accumulation of polyphenolic compounds and carbohydrates straining in Figure 2 should be separated into two parts of results, with Figure 2A providing polyphenolic compounds straining and Figure 2B presenting carbohydrates straining by comparing all treatments of grafted cucumber with control.
- Figure 3 should add A and B in Figures. The authors should explain the more detail in Figure 3 and add the number of sample and statistical analysis.
- Figure 4 has to be reorganized due to a lack of scientific presentation.
- Add A, B, C and D in each figure to make figure 4 A, B, C and D.
- Write the explanation of X and Y axis in all figures.
- The same treatment name should be used on the X axis as in the other figure results.
- Explain the more detail in Figure 4 and add the number of samples.
- It appears that the discussion looks similar to the introduction. More research and discussion into how the grafted cucumber can withstand illnesses and heat stress is required.
Author Response
Reviewer #3
Comments and Suggestions for Authors
According to the manuscript “Can Grafting Control Fusarium Wilt Disease of Cucumber and its Productivity under Heat Stress?” submitted in the section of plant molecular biology in special issue of control of plant pathogens for a greener future: induced systemic resistance and epigenetics of Plants journal. The authors report the response of grafted cucumber plants onto five cucurbit rootstocks infected with Fusarium oxysporum f. sp. cucumerinum alone and in combination with fusarium wilt disease and heat stress in two different locations in Egypt. The investigation's goal is self-evident. However, the concept is not novel and lack of information on systemic defense mechanisms. It has a critical flaw that must be fixed. Furthermore, the paper's structure must be restructured, and there are a number of issues to be concerned about.
- This research seems a lot like the one you did before “Grafting Improves Fruit Yield of Cucumber Plants Grown under Combined Heat and Soil Salinity Stresses”. To make the work valuable and intriguing to the reader, the author should conduct extra research. It is necessary to investigate the plant's systemic defense mechanisms such as expression of pathogen and stress related genes, defense-related enzymes, and defense-related phytohormones etc.
Response: In our lab, we are currently using grafting technology as applicable tool to improve fruit yield and quality in cucumber and tomato under both biotic and abiotic stress. In Bayoumi et al. (2021), our research focused on using grafting to improve cucumber productivity under combined abiotic stresses (heat and salinity). In the present study, we assess the response of grafted cucumber onto cucurbit rootstocks with more focus on horticultural traits under combined heat stress and fusarium wilt under greenhouse conditions. In addition, we study the anatomical structure as showed in Fig. 2 which resulted different compatibilities under disease pressure and heat stress.
We agree with your comment, other mechanisms such as expression of pathogen, stress related genes, defense-related enzymes, and defense-related phytohormones etc. will provide more information in terms of interactions between host and pathogen, but these tests are expensive and we do not have enough facilities for such these measurements.
- Visual data of disease severity and yield size after grafting should be added in order to clearly show the effect between control plants and cucumber grafted plants.
Response: Already done as shown in Table 1 and Fig. 1 for disease severity% and in Fig. 3 for total fruit yield (kg/plant), thanks!
- Table 1 and 2, the authors should explain the meaning of * or ** in the row of F-test and write the number of sample and replicate in each experiment.
Response: OK thanks, we added in the revised Manuscript in the capture of both tables
Number of samples was 20 plants as 4 plants per each replicate (4 plants × 5 replicates)
- The disease severity and efficiency percentage should divide into 2 figure like figure 1A is disease severity and figure 1B is the efficiency percentage and add the culture locations in same figure.
Response: OK Thanks, we added in Results section in the revised MS
- Figure 2 is hard to understand since it only shows the structure of VSS-61 F1 and Bottle gourd rootstocks. The accumulation of polyphenolic compounds and carbohydrates straining in Figure 2 should be separated into two parts of results, with Figure 2A providing polyphenolic compounds straining and Figure 2B presenting carbohydrates straining by comparing all treatments of grafted cucumber with control.
Response: We could not collect samples from control infected plants where all plants were dead very fast. The figure was divided into 4 groups based on your advice as in the revised MS in lines from 170 to 185 and a figure for all steps of grafting process also was added in the revised MS
- Figure 3 should add A and B in Figures. The authors should explain the more detail in Figure 3 and add the number of sample and statistical analysis.
Response: OK, Thanks, Done
- Figure 4 has to be reorganized due to a lack of scientific presentation.
- Add A, B, C and D in each figure to make figure 4 A, B, C and D
- Response: OK, Thanks, Done
- Write the explanation of X and Y axis in all figures.
- Response: OK, Thanks, Done
- The same treatment name should be used on the X axis as in the other figure results.
- Response: OK, Thanks, Done
- Explain the more detail in Figure 4 and add the number of samples.
Response: Number of samples was 15 fruits as 5 fruits per each replicate (5 fruits * 3 replicates)
- It appears that the discussion looks similar to the introduction. More research and discussion into how the grafted cucumber can withstand illnesses and heat stress is required.
Response: the section of discussion is improved to include many added parts concerning the pathogens, heat stress and some available published refs.
Also, a figure summarize different mechanism about the tolerance or resistance abiotic or biotic stress was added to revised MS.
Lines from 286 to 304, 321 to 329, 327 to 335, 346 to 31 and 359 to 374.

Round 2
Reviewer 3 Report
After reading the manuscript, the authors have a better edit. However, more experiments are needed to explain the importance of grafting to disease and heat resistance, which will make the research more interesting.
Best regards
Author Response
Reviewer #1
Comments and Suggestions for Authors
After reading the manuscript, the authors have a better edit. However, more experiments are needed to explain the importance of grafting to disease and heat resistance, which will make the research more interesting.
Response: added, thanks in the revised MS in line from 375 to 387.
